# Supervising the Multi-Fidelity Race
# of Hyperparameter Configurations

**Martin Wistuba**[*][†]
Amazon Web Services, Berlin, Germany
marwistu@amazon.com

**Arlind Kadra**[*]
University of Freiburg, Freiburg, Germany
kadraa@cs.uni-freiburg.de

**Josif Grabocka**
University of Freiburg, Freiburg, Germany
grabocka@cs.uni-freiburg.de

## Abstract

Multi-fidelity (gray-box) hyperparameter optimization techniques (HPO) have recently emerged as a promising direction for tuning Deep Learning methods. However, existing methods suffer from a sub-optimal allocation of the HPO budget to the hyperparameter configurations. In this work, we introduce DyHPO, a Bayesian Optimization method that learns to decide which hyperparameter configuration to train further in a dynamic race among all feasible configurations. We propose a new deep kernel for Gaussian Processes that embeds the learning curve dynamics, and an acquisition function that incorporates multi-budget information. We demonstrate the significant superiority of DyHPO against state-of-the-art hyperparameter optimization methods through large-scale experiments comprising 50 datasets (Tabular, Image, NLP) and diverse architectures (MLP, CNN/NAS, RNN).

## 1 Introduction

Hyperparameter Optimization (HPO) is arguably an acute open challenge for Deep Learning (DL), especially considering the crucial impact HPO has on achieving state-of-the-art empirical results. Unfortunately, HPO for DL is a relatively under-explored field and most DL researchers still optimize their hyperparameters via obscure trial-and-error practices. On the other hand, traditional Bayesian Optimization HPO methods [Snoek et al., 2012, Bergstra et al., 2011] are not directly applicable to deep networks, due to the infeasibility of evaluating a large number of hyperparameter configurations. In order to scale HPO for DL, three main directions of research have been recently explored. *(i) Online HPO* methods search for hyperparameters during the optimization process via meta-level controllers [Chen et al., 2017, Parker-Holder et al., 2020], however, this online adaptation can not accommodate all hyperparameters (e.g. related to architectural changes). *(ii) Gradient-based HPO* techniques, on the other hand, compute the derivative of the validation loss w.r.t. hyperparameters by reversing the training update steps [Maclaurin et al., 2015, Franceschi et al., 2017, Lorraine et al., 2020], however, the reversion is not directly applicable to all cases (e.g. dropout rate). The last direction, *(iii) Gray-box HPO* techniques discard sub-optimal configurations after evaluating them on lower budgets [Li et al., 2017, Falkner et al., 2018].

In contrast to the online and gradient-based alternatives, gray-box approaches can be deployed in an off-the-shelf manner to all types of hyperparameters and architectures. The gray-box concept is based on the intuition that a poorly-performing hyperparameter configuration can be identified and

---

[*]equal contribution

[†]work does not relate to position at Amazon

36th Conference on Neural Information Processing Systems (NeurIPS 2022).

terminated by inspecting the validation loss of the first few epochs, instead of waiting for the full convergence. The most prominent gray-box algorithm is Hyperband [Li et al., 2017], which is based on successive halving. It runs random configurations at different budgets (e.g. number of epochs) and successively halves these configurations by keeping only the top performers. Follow-up works, such as BOHB [Falkner et al., 2018] or DEHB [Awad et al., 2021], replace the random sampling of Hyperband with a sampling based on Bayesian optimization or differentiable evolution.

Despite their great practical potential, gray-box methods suffer from a major issue. The low-budget (few epochs) performances are not always a good indicator for the full-budget (full convergence) performances. For example, a properly regularized network converges slower in the first few epochs, however, typically performs better than a non-regularized variant after the full convergence. In other words, there can be a poor rank correlation of the configurations' performances at different budgets.

We introduce DYHPO, a Bayesian Optimization (BO) approach based on Gaussian Processes (GP), that proposes a novel treatment to the multi-budget (a.k.a. multi-fidelity) setup. In this perspective, we propose a deep kernel GP that captures the learning dynamics. As a result, we train a kernel capable of capturing the similarity of a pair of hyperparameter configurations, even if the pair's configurations are evaluated at different budgets. Furthermore, we extend Expected Improvement [Jones et al., 1998] to the multi-budget case, by introducing a new mechanism for the incumbent configuration of a budget.

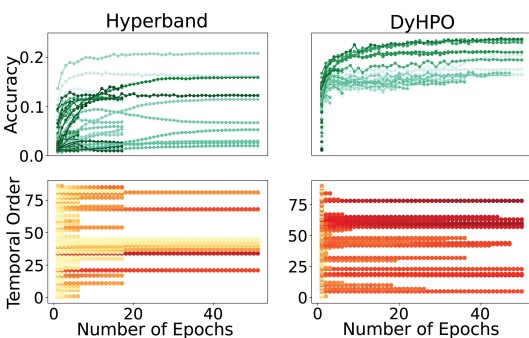

Figure 1: **Top:** The learning curve for different hyperparameter configurations. The darker the learning curve, the later it was evaluated during the search. **Bottom:** The hyperparameter indices in a temporal order as evaluated during the optimization and their corresponding curves.

We illustrate the differences between our racing strategy and successive halving with the experiment of Figure 1, where, we showcase the HPO progress of two different methods on the "Helena" dataset from the LCBench benchmark [Zimmer et al., 2021]. Hyperband [Li et al., 2017] is a gray-box approach that *statically* pre-allocates the budget for a set of candidates (Hyperband bracket) according to a predefined policy. However, DYHPO *dynamically* adapts the allocation of budgets for configurations after every HPO step (a.k.a. a dynamic race). As a result, DYHPO invests only a small budget on configurations that show little promise as indicated by the intermediate scores.

The joint effect of modeling a GP kernel across budgets together with a dedicated acquisition function leads to DYHPO achieving a statistically significant empirical gain against state-of-the-art gray-box baselines [Falkner et al., 2018, Awad et al., 2021], including prior work on multi-budget GPs [Kandasamy et al., 2017, 2020] or neural networks [Li et al., 2020b]. We demonstrate the performance of DYHPO in three diverse deep learning architectures (MLP, CNN/NAS, RNN) and 50 datasets of three diverse modalities (tabular, image, natural language processing). We believe our method is a step forward toward making HPO for DL practical and feasible. Overall, our contributions can be summarized as follows:

- We introduce a novel Bayesian surrogate for gray-box HPO optimization. Our novel surrogate model predicts the validation score of a machine learning model based on both the hyperparameter configuration, the budget information, and the learning curve.

- We derive a simple yet robust way to combine this surrogate model with Bayesian optimization, reusing most of the existing components currently used in traditional Bayesian optimization methods.

- Finally, we demonstrate the efficiency of our method for HPO and neural architecture search tasks compared to the current state-of-the-art methods in HPO, by outperforming seven strong HPO baselines with a statistically significant margin. As an overarching goal, we believe our method is an important step toward scaling HPO for DL.

## 2 Related Work on Gray-box HPO

**Multi-Fidelity Bayesian Optimization and Bandits.** Bayesian optimization is a black-box function optimization framework that has been successfully applied in optimizing hyperparameter and neural architectures alike [Snoek et al., 2012, Kandasamy et al., 2018, Bergstra et al., 2011]. To further improve Bayesian optimization, several works propose low-fidelity data approximations of hyperparameter configurations by training on a subset of the data [Swersky et al., 2013, Klein et al., 2017a], or by terminating training early [Swersky et al., 2014]. Additionally, several methods extend Bayesian optimization to multi-fidelity data by engineering new kernels suited for this problem [Swersky et al., 2013, 2014, Poloczek et al., 2017]. Kandasamy et al. [2016] extends GP-UCB [Srinivas et al., 2010] to the multi-fidelity setting by learning one Gaussian Process (GP) with a standard kernel for each fidelity. Their later work improves upon this method by learning one GP for all fidelities that enables the use of continuous fidelities [Kandasamy et al., 2017]. The work by Takeno et al. [2020] follows a similar idea but proposes to use an acquisition function based on information gain instead of UCB. While most of the works rely on GPs to model the surrogate function, Li et al. [2020b] use a Bayesian neural network that models the complex relationship between fidelities with stacked neural networks, one for each fidelity.

Hyperband [Li et al., 2017] is a bandits-based multi-fidelity method for hyperparameter optimization that selects hyperparameter configurations at random and uses successive halving [Jamieson and Talwalkar, 2016] with different settings to early-stop less promising training runs. Several improvements have been proposed to Hyperband with the aim to replace the random sampling of hyperparameter configurations with a more guided approach [Bertrand et al., 2017, Wang et al., 2018, Wistuba, 2017]. BOHB [Falkner et al., 2018] uses TPE [Bergstra et al., 2011] and builds a surrogate model for every fidelity adhering to a fixed-fidelity selection scheme. DEHB [Awad et al., 2021] samples candidates using differential evolution which handles large and discrete search spaces better than BOHB. Mendes et al. [2021] propose a variant of Hyperband which allows to skip stages.

**Learning Curve Prediction** A variety of methods attempt to extrapolate a partially observed learning curve in order to estimate the probability that a configuration will improve over the current best solution. Domhan et al. [2015] propose to ensemble a set of parametric functions to extrapolate a partial learning curve. While this method is able to extrapolate with a single example, it requires a relatively long learning curve to do so. The work by Klein et al. [2017b] build upon the idea of using a set of parametric functions. The main difference is that they use a heteroscedastic Bayesian model to learn the ensemble weights. Baker et al. [2018] propose to use support vector machines (SVM) as an auto-regressive model. The SVM predicts the next value of a learning curve, the original learning curve is augmented by this value and we keep predicting further values. The work by Gargiani et al. [2019] use a similar idea but makes prediction based on the last $K$ observations only and uses probabilistic models. Wistuba and Pedapati [2020] propose to learn a prediction model across learning curves from different tasks to avoid the costly learning curve collection. In contrast to DYHPO, none of these methods selects configuration but is limited to deciding when to stop a running configuration.

**Multi-Fidelity Acquisition Functions** Klein et al. [2017a] propose an acquisition function which allows for selecting hyperparameter configurations and the dataset subset size. The idea is to reduce training time by considering only a smaller part of the training data. In contrast to $\text{EI}_{\text{MF}}$, this acquisition function is designed to select arbitrary subset sizes whereas $\text{EI}_{\text{MF}}$ is intended to slowly increase the invested budget over time. Mendes et al. [2020] extend the work of Klein et al. [2017a] to take business constraints into account.

**Deep Kernel Learning with Bayesian Optimization.** We are among the first to use deep kernel learning with Bayesian optimization and to the best of our knowledge the first to use it for multi-fidelity Bayesian optimization. Rai et al. [2016] consider the use of a deep kernel instead of a manually designed kernel in the context of standard Bayesian optimization, but, limit their experimentation to synthetic data and do not consider its use for hyperparameter optimization. Perrone et al. [2018], Wistuba and Grabocka [2021] use a pre-trained deep kernel to warm start Bayesian optimization with meta-data from previous optimizations. The aforementioned approaches are multi-task or transfer learning methods that require the availability of meta-data from related tasks.

In contrast to prior work, we propose a method that introduces deep learning to multi-fidelity HPO with Bayesian Optimization, and captures the learning dynamics across fidelities/budgets, combined with an acquisition function that is tailored for the gray-box setup.

## 3 Dynamic Multi-Fidelity HPO

### 3.1 Preliminaries

**Gray-Box Optimization.** The gray-box HPO setting allows querying configurations with a smaller budget compared to the total maximal budget $B$. Thus, we can query from the response function $f : \mathcal{X} \times \mathbb{N} \to \mathbb{R}$ where $f_{i,j} = f(\mathbf{x}_i, j)$ is the response after spending a budget of $j$ on configuration $\mathbf{x}_i$. As before, these observations are noisy and we observe $y_{i,j} = f(\mathbf{x}_i, j) + \varepsilon_j$ where $\varepsilon_j \sim \mathcal{N}(0, \sigma_{j,n}^2)$. Please note, we assume that the budget required to query $f_{i,j+b}$ after having queried $f_{i,j}$ is only $b$. Furthermore, we use the learning curve $\mathbf{Y}_{i,j-1} = (y_{i,1}, \ldots, y_{i,j-1})$ when predicting $f_{i,j}$.

**Gaussian Processes (GP).** Given a training data set $\mathcal{D} = \{(\mathbf{x}_i, y_i)\}_{i=1}^n$, the Gaussian Process assumption is that $y_i$ is a random variable and the joint distribution of all $y_i$ is assumed to be multivariate Gaussian distributed as $\mathbf{y} \sim \mathcal{N}(m(\mathbf{X}), k(\mathbf{X}, \mathbf{X}))$. Furthermore, $\mathbf{f}_*$ for test instances $\mathbf{x}_*$ are jointly Gaussian with $\mathbf{y}$ as:

$$\begin{bmatrix} \mathbf{y} \\ \mathbf{f}_* \end{bmatrix} \sim \mathcal{N} \left( m(\mathbf{X}, \mathbf{x}_*), \begin{pmatrix} \mathbf{K}_n & \mathbf{K}_* \\ \mathbf{K}_*^T & \mathbf{K}_{**} \end{pmatrix} \right). \tag{1}$$

The mean function $m$ is often set to $\mathbf{0}$ and its covariance function $k$ depends on parameters $\boldsymbol{\theta}$. For notational convenience, we use $\mathbf{K}_n = k(\mathbf{X}, \mathbf{X}|\boldsymbol{\theta}) + \sigma_n^2 \mathbf{I}$, $\mathbf{K}_* = k(\mathbf{X}, \mathbf{X}_*|\boldsymbol{\theta})$ and $\mathbf{K}_{**} = k(\mathbf{X}_*, \mathbf{X}_*|\boldsymbol{\theta})$ to define the kernel matrices. We can derive the posterior predictive distribution with mean and covariance as follows:

$$\mathbb{E}[\mathbf{f}_*|\mathbf{X}, \mathbf{y}, \mathbf{X}_*] = \mathbf{K}_*^T \mathbf{K}_n^{-1} \mathbf{y}, \ \mathrm{cov}[\mathbf{f}_*|\mathbf{X}, \mathbf{X}_*] = \mathbf{K}_{**} - \mathbf{K}_*^T \mathbf{K}_n^{-1} \mathbf{K}_* \tag{2}$$

Often, the kernel function is manually engineered, one popular example is the squared exponential kernel. However, in this work, we make use of the idea of deep kernel learning [Wilson et al., 2016]. The idea is to model the kernel as a neural network $\varphi$ and learn the best kernel transformation $\mathbf{K}(\theta, w) := k(\varphi(\mathbf{x}, w), \varphi(\mathbf{x}'; w)|\boldsymbol{\theta})$, which allows us to use convolutional operations in our kernel.

### 3.2 Deep Multi-Fidelity Surrogate

We propose to use a Gaussian Process surrogate model that infers the value of $f_{i,j}$ based on the hyperparameter configuration $\mathbf{x}_i$, the budget $j$ as well as the past learning curve $\mathbf{Y}_{i,j-1}$. For this purpose, we use a deep kernel as:

$$\mathbf{K}(\theta, w) := k(\varphi(\mathbf{x}_i, \mathbf{Y}_{i,j-1}, j; w), \varphi(\mathbf{x}_{i'}, \mathbf{Y}_{i',j'-1}, j'; w); \theta) \tag{3}$$

We use a squared exponential kernel for $k$ and the neural network $\varphi$ is composed of linear and convolutional layers as shown in Figure 2. We normalize the budget $j$ to a range between 0 and 1 by dividing it by the maximum budget $B$. Afterward, it is concatenated with the hyperparameter configuration $\mathbf{x}_i$ and fed to a linear layer. The learning curve $\mathbf{Y}_{i,j-1}$ is transformed by a one-dimensional convolution followed by a global max pooling layer. Finally, both representations are fed to another linear layer.

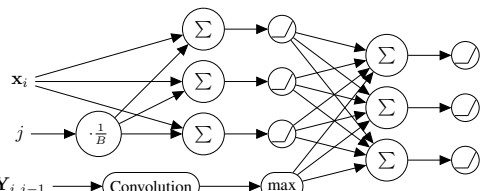

Figure 2: The feature extractor $\varphi$ of our kernel.

Its output will be the input to the kernel function $k$. Both, the kernel $k$ and the neural network $\varphi$ consist of trainable parameters $\boldsymbol{\theta}$ and $\mathbf{w}$, respectively. We find their optimal values by computing the maximum likelihood estimates as:

$$\hat{\theta}, \hat{w} = \underset{\theta, w}{\arg\max}\, p(\mathbf{y}|\mathbf{X}, \mathbf{Y}, \theta, w) \propto \underset{\theta, w}{\arg\min}\, \mathbf{y}^{\mathrm{T}} \mathbf{K}(\theta, w)^{-1} \mathbf{y} + \log|\mathbf{K}(\theta, w)| \tag{4}$$

In order to solve this optimization problem, we use gradient descent and Adam [Kingma and Ba, 2015] with a learning rate of $0.1$. Given the maximum likelihood estimates, we can approximate the predictive posterior through $p\left(f_{i,j}|\mathbf{x}_i, \mathbf{Y}_{i,j-1}, j, \mathcal{D}, \hat{\boldsymbol{\theta}}, \hat{\mathbf{w}}\right)$, and ultimately compute the mean and covariance of this Gaussian using Equation 2.

### 3.3 Multi-Fidelity Expected Improvement

Expected improvement [Jones et al., 1998] is a commonly used acquisition function and is defined as:

$$\mathrm{EI}(\mathbf{x}|\mathcal{D}) = \mathbb{E}\left[\max\left\{f(\mathbf{x}) - y^{\mathrm{max}}, 0\right\}\right] , \tag{5}$$

where $y^{\mathrm{max}}$ is the largest observed value of $f$. We propose a multi-fidelity version of it as:

$$\mathrm{EI}_{\mathrm{MF}}(\mathbf{x}, j|\mathcal{D}) = \mathbb{E}\left[\max\left\{f(\mathbf{x}, j) - y_j^{\mathrm{max}}, 0\right\}\right] , \tag{6}$$

where:

$$y_j^{\mathrm{max}} = \begin{cases} \max\left\{y \mid ((\mathbf{x}, \cdot, j), y) \in \mathcal{D}\right\} & \text{if } ((\mathbf{x}, \cdot, j), y) \in \mathcal{D} \\ \max\left\{y \mid (\cdot, y) \in \mathcal{D}\right\} & \text{otherwise} \end{cases} \tag{7}$$

Simply put, $y_j^{\mathrm{max}}$ is the largest observed value of $f$ for a budget of $j$ if it exists already, otherwise, it is the largest observed value for any budget. If there is only one possible budget, the multi-fidelity expected improvement is identical to expected improvement.

### 3.4 The DYHPO Algorithm

The DYHPO algorithm looks very similar to many black-box Bayesian optimization algorithms as shown in Algorithm 1. The big difference is that at each step we dynamically decide which candidate configuration to train *for a small additional budget*.

Possible candidates are previously unconsidered configurations as well as configurations that did not reach the maximum budget. In Line 2, the most promising candidate is chosen using the acquisition function introduced in Section 3.3 and the surrogate model's predictions. It is important to highlight that we do not maximize the acquisition function along the budget dimensionality. Instead, we set the budget $b$ such that it is by exactly one higher than the budget used to evaluate $\mathbf{x}_i$ before. This ensures that we

---

**Algorithm 1** DYHPO Algorithm

---
1: $b(\mathbf{x}) = 0 \; \forall \mathbf{x} \in \mathcal{X}$
2: **while** not converged **do**
3:    $\mathbf{x}_i \leftarrow \arg\max_{\mathbf{x} \in \mathcal{X}} \mathrm{EI}_{\mathrm{MF}}\left(\mathbf{x}, b(\mathbf{x}) + 1\right)$ (Sec. 3.3)
4:    Observe $y_{i,b(\mathbf{x}_i)+1}$.
5:    $b(\mathbf{x}_i) \leftarrow b(\mathbf{x}_i) + 1$
6:    $\mathcal{D} \leftarrow \mathcal{D} \cup \left\{\left((\mathbf{x}_i, \mathbf{Y}_{i,b(\mathbf{x}_i)-1}, b(\mathbf{x}_i)), y_{i,b(\mathbf{x}_i)}\right)\right\}$
7:    Update the surrogate on $\mathcal{D}$. (Sec. 3.2)
   **return** $\mathbf{x}_i$ with largest $y_{i,\cdot}$.

---

explore configurations by slowly increasing the budget. After the candidate and the corresponding budget are selected, the function $f$ is evaluated and we observe $y_{i,j}$ (Line 3). This additional data point is added to $\mathcal{D}$ in Line 4. Then in Line 5, the surrogate model is updated according to the training scheme described in Section 3.2.

## 4 Experimental Protocol

### 4.1 Experimental Setup

We evaluate DYHPO in three different settings on hyperparameter optimization for tabular, text, and image classification against several competitor methods, the details of which are provided in the following subsections. We ran all of our experiments on an Amazon EC2 M5 Instance (m5.xlarge). In our experiments, we report the mean of ten repetitions and we report two common metrics, the regret and the average rank. The regret refers to the absolute difference between the score of the solution found by an optimizer compared to the best possible score. If we report the regret as an aggregate result over multiple datasets, we report the mean over all regrets. The average rank is the

metric we use to aggregate rank results over different datasets. We provide further implementation and training details in Appendix A.4. Our implementation of DYHPO is publicly available.[3]

## 4.2 Benchmarks

In our experiments, we use the following benchmarks. We provide more details in Appendix A.1.

**LCBench:** A learning curve benchmark [Zimmer et al., 2021] that evaluates neural network architectures for tabular datasets. LCBench contains learning curves for 35 different datasets, where 2,000 neural networks per dataset are trained for 50 epochs with Auto-PyTorch.

**TaskSet:** A benchmark that features diverse tasks Metz et al. [2020] from different domains and includes 5 search spaces with different degrees of freedom, where, every search space includes 1000 hyperparameter configurations. In this work, we focus on a subset of NLP tasks (12 tasks) and we use the Adam8p search space with 8 continuous hyperparameters.

**NAS-Bench-201:** A benchmark consisting of 15625 hyperparameter configurations representing different architectures on the CIFAR-10, CIFAR-100 and ImageNet datasets Dong and Yang [2020]. NAS-Bench-201 features a search space of 6 categorical hyperparameters and each architecture is trained for 200 epochs.

## 4.3 Baselines

**Random Search:** A random/stochastic black-box search method for HPO.

**HyperBand:** A multi-arm bandit method that extends successive halving by multiple brackets with different combinations of the initial number of configurations, and their initial budget [Li et al., 2017].

**BOHB:** An extension of Hyperband that replaces the random sampling of the initial configurations for each bracket with recommended configurations from a model-based approach [Falkner et al., 2018]. BOHB builds a model for every fidelity that is considered.

**DEHB:** A method that builds upon Hyperband by exploiting differential evolution to sample the initial candidates of a Hyperband bracket [Awad et al., 2021].

**ASHA:** An asynchronous version of successive halving (or an asynchronous version of Hyperband if multiple brackets are run). ASHA Li et al. [2020a] does not wait for all configurations to finish inside a successive halving bracket, but, instead promotes configurations to the next successive halving bracket in real-time.

**MF-DNN:** A multi-fidelity Bayesian optimization method that uses deep neural networks to capture the relationships between different fidelities Li et al. [2020b].

**Dragonfly:** We compare against BOCA [Kandasamy et al., 2017] by using the Dragonfly library Kandasamy et al. [2020]. This method suggests the next hyperparameter configuration as well as the budget it should be evaluated for.

## 4.4 Research Hypotheses and Associated Experiments

**Hypothesis 1:** DYHPO achieves state-of-the-art results in multi-fidelity HPO.

**Experiment 1:** We compare DYHPO against the baselines of Section 4.3 on the benchmarks of Section 4.2 with the experimental setup of Section 4.1. For TaskSet we follow the authors' recommendation and report the number of steps (every 200 iterations).

**Hypothesis 2:** DYHPO's runtime overhead has a negligible impact on the quality of results.

**Experiment 2:** We compare DYHPO against the baselines of Section 4.3 over the wallclock time. The wallclock time includes both *(i)* the optimizer's runtime overhead for recommending the next hyperparameter configuration, plus *(ii)* the time needed to evaluate the recommended configuration. In this experiment, we consider all datasets where the average training time per epoch is at least 10 seconds, because, for tasks where the training time is short, there is no practical justification for

---

[3]`https://github.com/releaunifreiburg/DyHPO`

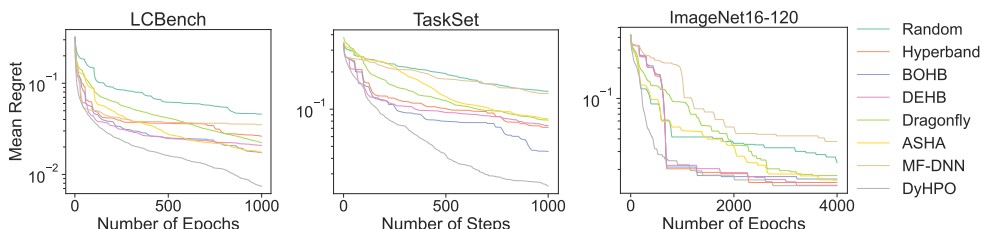

Figure 3: The mean regret for the different benchmarks over the number of epochs or steps (every 200 iterations). The results are aggregated over 35 different datasets for LCBench and aggregated over 12 different NLP tasks for TaskSet.

complex solutions and their overhead. In these cases, we recommend using a random search. We don't report results for TaskSet because the benchmark lacks training times.

**Hypothesis 3:** DYHPO uses the computational budget more efficiently than baselines.

**Experiment 3:** To further verify that DYHPO is efficient compared to the baselines, we investigate whether competing methods spend their budgets on qualitative candidates. Concretely we: i) calculate the precision of the top (w.r.t. ground truth) performing configurations that were selected by each method across different budgets, ii) compute the average regret of the selected configurations across budget, and iii) we compare the fraction of top-performing configurations at a given budget that were not top performers at lower budgets, i.e. measure the ability to handle the poor correlation of performances across budgets.

# 5 Results

**Experiment 1: DYHPO achieves state-of-the-art results.** In our first experiment, we evaluate the various methods on the benchmarks listed in Section 4.2. We show the aggregated results in Figure 3, the results show that DYHPO manages to outperform competitor methods over the set of considered benchmarks by achieving a better mean regret across datasets. Not only does DYHPO achieve a better final performance, it also achieves strong anytime results by converging faster than the competitor methods. For the extended results, related to the performance of all methods on a dataset level, we refer the reader to Appendix B.



Figure 4: Critical difference diagram for LCBench and TaskSet in terms of the number of HPO steps. The results correspond to results after 500 and 1,000 epochs. Connected ranks via a bold bar indicate that performances are not significantly different ($p > 0.05$).

In Figure 4, we provide further evidence that DYHPO's improvement over the baselines is statistically significant. The critical difference diagram presents the ranks of all methods and provides information on the pairwise statistical difference between all methods for two fractions of the number of HPO steps (50% and 100%). We included the LCBench and TaskSet benchmarks in our significance plots. NAS-Bench-201 was omitted because it has only 3 datasets and the statistical test cannot be applied. Horizontal lines indicate groupings of methods that are not significantly different. As suggested by the best published practices Demsar [2006], we use the Friedman test to reject the null hypothesis followed by a pairwise post-hoc analysis based on the Wilcoxon signed-rank test ($\alpha = 0.05$).

For LCBench, DYHPO already outperforms the baselines significantly after 50% of the search budget, with a statistically significant margin. As the optimization procedure continues, DYHPO manages to extend its gain in performance and is the only method that has a statistically significant improvement against all the other competitor methods. Similarly, for TaskSet, DYHPO manages to outperform all methods with a statistically significant margin only halfway through the optimization procedure and achieves the best rank over all methods. However, as the optimization procedure continues, BOHB

manages to decrease the performance gap with DYHPO, although, it still achieves a worse rank across all datasets. Considering the empirical results, we conclude that **Hypothesis 1 is validated and that DYHPO achieves state-of-the-art results on multi-fidelity HPO**.

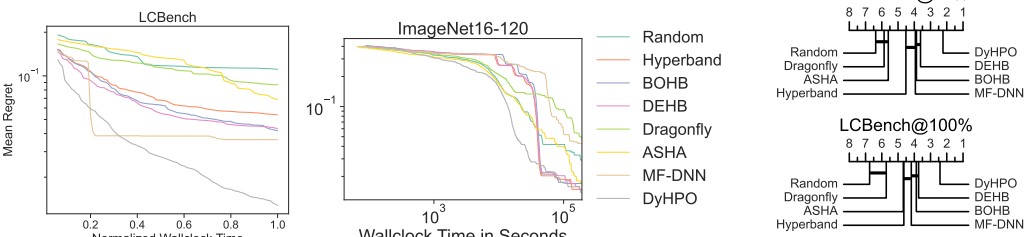

Figure 5: **Left:** The regret over time for all methods during the optimization procedure for the LCBench benchmark and the ImageNet dataset from the NAS-Bench-201 benchmark. The normalized wall clock time represents the actual run time divided by the total wall clock time of DYHPO including the overhead of fitting the deep GP. **Right:** The critical difference diagram for LCBench halfway through the HPO wall-clock time, and in the end. Connected ranks via a bold bar indicate that differences are not significant ($p > 0.05$).

**Experiment 2: On the impact of DYHPO's overhead on the results.** We present the results of our second experiment in Figure 5 (left), where, as it can be seen, DYHPO still outperforms the other methods when its overhead is considered. For LCBench, DYHPO manages to get an advantage fairly quickly and it only increases the gap in performance with the other methods as the optimization process progresses. Similarly, in the case of ImageNet from NAS-Bench-201, DYHPO manages to gain an advantage earlier than other methods during the optimization procedure. Although in the end DYHPO still performs better than all the other methods, we believe most of the methods converge to a good solution and the differences in the final performance are negligible. For the extended results, related to the performance of all methods on a dataset level over time, we refer the reader to the plots in Appendix B. Additionally, in Figure 5 (right), we provide the critical difference diagrams for LCBench that present the ranks and the statistical difference of all methods halfway through the optimization procedure, and in the end. As it can be seen, DYHPO has a better rank with a significant margin with only half of the budget used and it retains the advantage until the end.

**Experiment 3: On the efficiency of DYHPO.** In Figure 6 (left), we plot the precision of every method for different budgets during the optimization procedure, which demonstrates that DYHPO effectively explores the search space and identifies promising candidates. The precision at an epoch $i$ is defined as the number of top 1% candidates that are trained, divided by the number of all candidates trained, both trained for at least $i$ epochs. The higher the precision, the more relevant candidates were considered and the less computational resources were wasted. For small budgets, the precision is low since DYHPO spends budget to consider various candidates, but then, promising candidates are successfully identified and the precision quickly increases. This argument is further supported in Figure 6 (middle), where we visualize the average regret of all the candidates trained for at least the specified number of epochs on the x-axis. In contrast to the regret plots, here we do not show the regret of the best configuration, but the mean regret of all the selected configurations. The analysis deduces a similar finding, our method DYHPO selects more qualitative hyperparameter configurations than all the baselines.

An interesting property of multi-fidelity HPO is the phenomenon of poor rank correlations among the validation performance of candidates at different budgets. In other words, a configuration that achieves a poor performance at a small budget can perform better at a larger budget. To analyze this phenomenon, we measure the percentage of "good" configurations at a particular budget, that were "bad" performers in at least one of the smaller budgets. We define a "good" performance at a budget B when a configuration achieves a validation accuracy ranked among the top 1/3 of the validation accuracies belonging to all the other configurations that were run until that budget B.

In Figure 6 (right), we analyze the percentage of "good" configurations at each budget denoted by the x-axis, that were "bad" performers in at least one of the lower budgets. Such a metric is a proxy for the degree of the promotion of "bad" configurations towards higher budgets. We present the analysis for all the competing methods of our experimental protocol from Section 4. We have additionally included the ground-truth line annotated as "Baseline", which represents the fraction of past poor

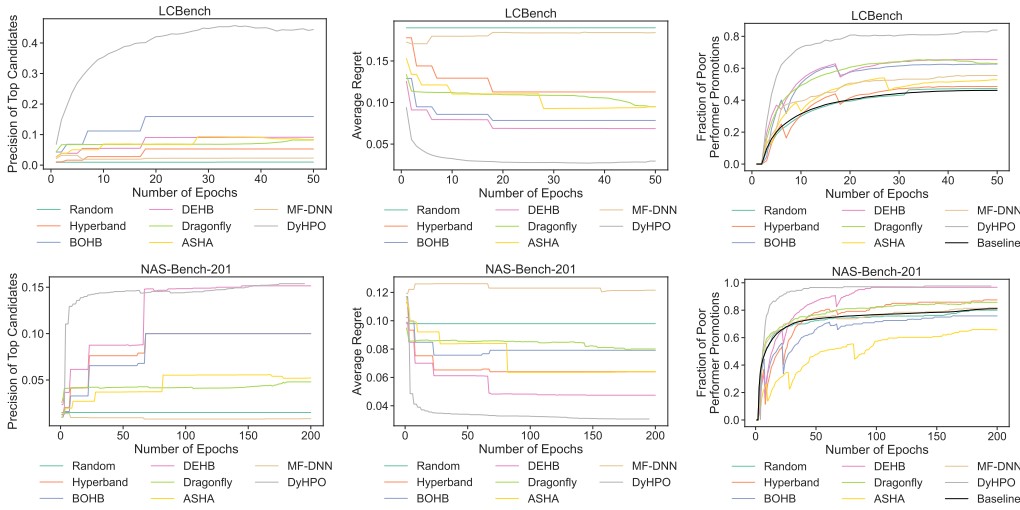

Figure 6: The efficiency of DYHPO as the optimization progresses. **Left:** The fraction of top-performing candidates from all candidates that were selected to be trained. **Middle:** The average regret for the configurations that were selected to be trained at a given budget. **Right:** The percentage of configurations that belong to the top 1/3 configurations at a given budget and that were in the top bottom 2/3 of the configurations at a previous budget. All of the results are from the LCBench and NAS-Bench-201 benchmark.

performers among all the feasible configurations in the search space. In contrast, the respective methods compute the fraction of promotions only among the configurations that those methods have considered (i.e. selected within their HPO trials) until the budget indicated by the x-axis. We see that there is a high degree of "good" configurations that were "bad" at a previous budget, with fractions of the ground-truth "Baseline" going up to 40% for the LCBench benchmark and up to 80% for the NAS-Bench-201 benchmark.

On the other hand, the analysis demonstrates that our method DYHPO has promoted more "good" configurations that were "bad" in a lower budget, compared to all the rival methods. In particular, more than 80% of selected configurations from the datasets belonging to either benchmark were "bad" performers at a lower budget. The empirical evidence validates **Hypothesis 3 and demonstrates that DYHPO efficiently explores qualitative candidates.** We provide the results of our analysis for DYHPO's efficiency on the additional benchmarks (Taskset) in Appendix B.

**Ablating the impact of the learning curve**

One of the main differences between DYHPO and similar methods Kandasamy et al. [2017], is that the learning curve is an input to the kernel function. For this reason, we investigate the impact of this design choice. We consider a variation of DYHPO w/o CNN, which is simply DYHPO without the learning curve.

It is worth emphasizing that both variants (with and without the learning curve) are multi-fidelity surrogates and both receive the budget information through the inputted index $j$ in Equation 3. The only difference is that DYHPO additionally incorporates the pattern of the learning curve.

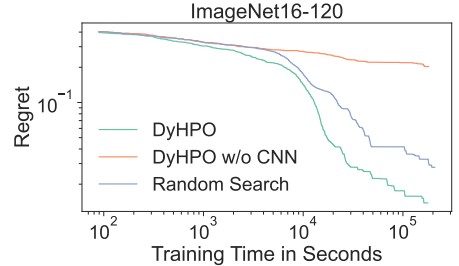

Figure 7: Ablating the impact of the learning curve on DYHPO.

We run the ablation on the NAS-Bench-201 benchmark and report the results for ImageNet, the largest dataset in our collection. The ablation results are shown in Figure 7, while the remaining results on the other datasets are shown in Figure 8 of the appendix. Based on the results from our

learning curve ablation, we conclude that the use of an explicit learning curve representation leads to significantly better results.

## 6    Limitations of Our Method

Although DYHPO shows a convincing and statistically significant reduction of the HPO time on diverse Deep Learning (DL) experiments, we cautiously characterized our method only as a "step towards" scaling HPO for DL. The reason for our restrain is the lack of tabular benchmarks for HPO on very large deep learning models, such as Transformers-based architectures [Devlin et al., 2019]. Additionally, the pause and resume part of our training procedure can only be applied when tuning the hyperparameters of parametric models, otherwise, the training of a hyperparameter configuration would have to be restarted. Lastly, for small datasets that can be trained fast, the overhead of model-based techniques would make an approach like random search more appealing.

## 7    Conclusions

In this work, we present DYHPO, a new Bayesian optimization (BO) algorithm for the gray-box setting. We introduced a new surrogate model for BO that uses a learnable deep kernel and takes the learning curve as an explicit input. Furthermore, we motivated a variation of expected improvement for the multi-fidelity setting. Finally, we compared our approach on diverse benchmarks on a total of 50 different tasks against the current state-of-the-art methods on gray-box hyperparameter optimization (HPO). Our method shows significant gains and has the potential to become the de facto standard for HPO in Deep Learning.

## Acknowledgments

Josif Grabocka and Arlind Kadra would like to acknowledge the grant awarded by the Eva-Mayr-Stihl Stiftung. In addition, this research was funded by the Deutsche Forschungsgemeinschaft (DFG, German Research Foundation) under grant number 417962828 and grant INST 39/963-1 FUGG (bwForCluster NEMO). In addition, Josif Grabocka acknowledges the support of the BrainLinks-BrainTools center of excellence.

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
