## Societal Implications

In our work, we use only publicly available data with no privacy concerns. Furthermore, our algorithm reduces the overall time for fitting deep networks, therefore, saving computational resources and yielding a positive impact on the environment. Moreover, our method can help smaller research organizations with limited access to resources to be competitive in the deep learning domain, which reduces the investment costs on hardware. Although our method significantly reduces the time taken for optimizing a machine learning algorithm that achieves peak performance, we warn against running our method for an extended time only to achieve marginal gains in performance, unless it is mission-critical. Last but not least, in order to save energy, we invite the community to create sparse benchmarks with surrogates, instead of dense tabular ones.

## A    Experimental Setup

### A.1    Benchmarks

**LCBench.**    LCBench[4] is a feedforward neural network benchmark on tabular data which consists of 2000 configuration settings for each of the 35 datasets. The configurations were evaluated during HPO runs with AutoPyTorch. LCBench features a search space of 7 numerical hyperparameters, where every hyperparameter configuration is trained for 50 epochs. The objective is to optimize seven different hyperparameters of funnel-shaped neural networks, i.e., batch size, learning rate, momentum, weight decay, dropout, number of layers, and maximum number of units per layer.

**TaskSet.**    TaskSet[5] is a benchmark that features over 1162 diverse tasks from different domains and includes 5 search spaces. In this work, we focus on NLP tasks and we use the Adam8p search space with 8 continuous hyperparameters. We refer to Figure 11 for the exact task names considered in our experiments. The learning curves provided in TaskSet report scores after every 200 iterations. We refer to those as "steps". The objective is to optimize eight hyperparameters for a set of different recurrent neural networks (RNN) that differ in embedding size, RNN cell, and other architectural features. The set of hyperparameters consists of optimizer-specific hyperparameters, such as the learning rate, the exponential decay rate of the first and second momentum of Adam, $\beta_1$ and $\beta_2$, and Adam's constant for numerical stability $\varepsilon$. Furthermore, there are two hyperparameters controlling linear and exponential learning rate decays, as well as L1 and L2 regularization terms.

**NAS-Bench-201.**    NAS-Bench-201[6] is a benchmark that has precomputed about 15,600 architectures trained for 200 epochs for the image classification datasets CIFAR-10, CIFAR-100, and ImageNet. The objective is to select for each of the six operations within the cell of the macro architecture one of five different operations. All other hyperparameters such as learning rate and batch size are kept fixed. NAS-Bench-201 features a search space of 6 categorical hyperparameters and each architecture is trained for 200 epochs.

### A.2    Preprocessing

In the following, we describe the preprocessing applied to the hyperparameter representation. For LCBench, we apply a log-transform to batch size, learning rate, and weight decay. For TaskSet, we apply it on the learning rate, L1 and L2 regularization terms, epsilon, linear and exponential decay of the learning rate. All continuous hyperparameters are scaled to the range between 0 and 1 using sklearn's MinMaxScaler. If not mentioned otherwise, we use one-hot encoding for the categorical hyperparameters. As detailed in subsection A.5, some baselines have a specific way of dealing with them. In that case, we use the method recommended by the authors.

### A.3    Framework

The framework contains the evaluated hyperparameters and their corresponding validation curves. The list of candidate hyperparameters is passed to the baseline-specific interface, which in turn,

---

[4]`https://github.com/automl/LCBench`
[5]`https://github.com/google-research/google-research/tree/master/task_set`
[6]`https://github.com/D-X-Y/NAS-Bench-201`

optimizes and queries the framework for the hyperparameter configuration that maximizes utility. Our framework in turn responds with the validation curve and the cost of the evaluation. In case a hyperparameter configuration has been evaluated previously up to a budget $b$ and a baseline requires the response for budget $b + 1$, the cost is calculated accordingly only for the extra budget requested.

### A.4 Implementation Details

We implement the Deep Kernel Gaussian Process using GPyTorch 1.5 [Gardner et al., 2018]. We use an RBF kernel and the dense layers of the transformation function $\varphi$ have 128 and 256 units. We used a convolutional layer with a kernel size of three and four filters. All parameters of the Deep Kernel are estimated by maximizing the marginal likelihood. We achieve this by using gradient ascent and Adam [Kingma and Ba, 2015] with a learning rate of 0.1 and batch size of 64. We stop training as soon as the training likelihood is not improving for 10 epochs in a row or we completed 1,000 epochs. For every new data point, we start training the GP with its old parameters to reduce the required effort for training.

### A.5 Baselines

**Random Search & Hyperband.** Random search and Hyperband sample hyperparameter configurations at random and therefore the preprocessing is irrelevant. We have implemented both from scratch and use the recommended hyperparameters for Hyperband, i.e. $\eta = 3$.

**BOHB.** For our experiments with BOHB, we use version 0.7.4 of the officially-released code[7].

**DEHB.** For our experiments with DEHB, we use the official public implementation[8]. We developed an interface that communicates between our framework and DEHB. In addition to the initial preprocessing common for all methods, we encode categorical hyperparameters with a numerical value in the interval [0, 1]. For a categorical hyperparameter $\mathbf{x}_i$, we take $N_i$ equal-sized intervals, where $N_i$ represents the number of unique categorical values for hyperparameter $\mathbf{x}_i$ and we assign the value for a categorical value $n \in N_i$ to the middle of the interval $[n, n + 1]$ as suggested by the authors. For configuring the DEHB algorithm we used the default values from the library.

**Dragonfly.** We use the publicly available code of Dragonfly[9]. No special treatment of categorical hyperparameters is required since Dragonfly has its own way to deal with them. We use version 0.1.6 with default settings.

**MF-DNN.** We use the official implementation of MF-DNN by the authors[10]. Initially, we tried to use multiple incremental fidelity levels like for DYHPO, however, the method runtime was too high and it could not achieve competitive results. For that reason, we use only a few fidelity levels like the authors do in their work Li et al. [2020b]. We use the same fidelity levels as for Hyperband, DEHB, and BOHB to have a fair comparison between the baselines. We also use the same number of initial points as for the other methods to have the same maximal resource allocated for every fidelity level.

**ASHA-HB.** We use the public implementation from the well-known optuna library (version 2.10.0). We used the same eta, minimum and maximal budget as for HB, DEHB, and BOHB in our experiments, to have a fair comparison.

## B  Additional Plots

In Figure 8, we ablate the learning curve input in our kernel, to see the effect it has on performance for the CIFAR-10 and CIFAR-100 datasets from the NAS-Bench-201 benchmark. The results indicate that the learning curve plays an important role in achieving better results by allowing faster convergence and a better anytime performance.

---

[7]`https://github.com/automl/HpBandSter`
[8]`https://github.com/automl/DEHB/`
[9]`https://github.com/dragonfly/dragonfly`
[10]`https://github.com/shib0li/DNN-MFBO`

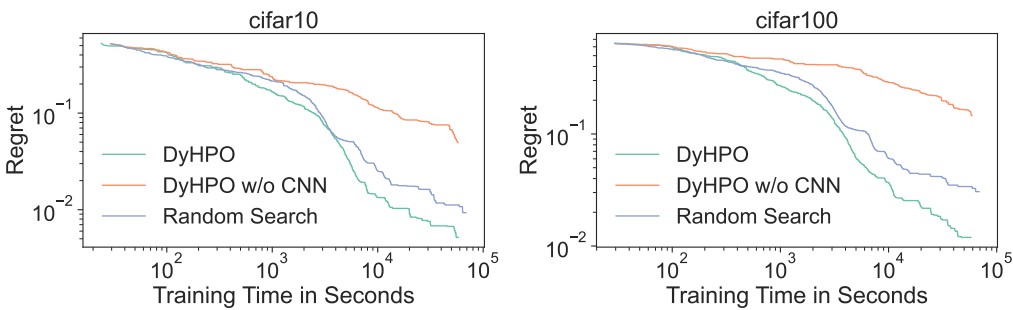

Figure 8: The learning curve ablation for the CIFAR-10 and CIFAR-100 tasks of NAS-Bench-201.

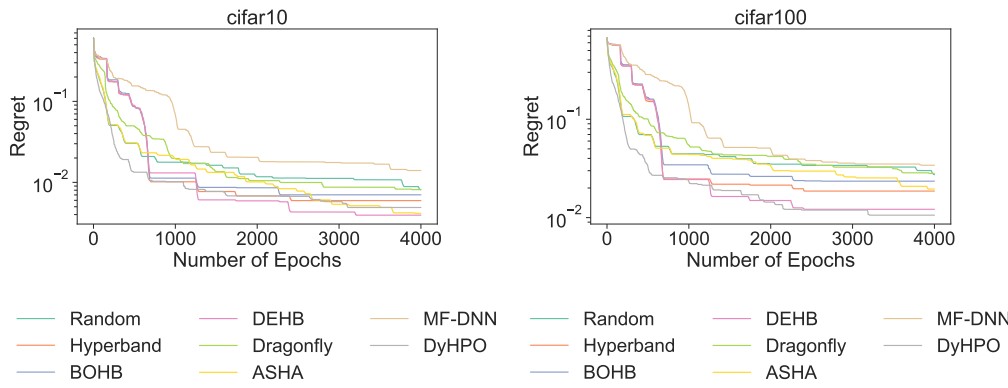

Figure 9: NAS-Bench-201 regret results over the number of epochs spent during the optimization.

Additionally, in Figure 9, we show the performance comparison over the number of epochs of every method for the CIFAR-10 and CIFAR-100 datasets in the NAS-Bench-201 benchmark. While, in Figure 10, we present the performance comparison over time. As can be seen, DYHPO converges faster and has a better performance compared to the other methods over the majority of the time or steps, however, towards the end although it is the optimal method or close to the optimal method, the difference in regret is not significant anymore.

Furthermore, Figure 11 shows the performance comparison for the datasets chosen from TaskSet over the number of steps. Looking at the results, DYHPO is outperforming all methods convincingly on the majority of datasets by converging faster and with significant differences in the regret evaluation metric.

In Figure 12 and 13, we show the performance comparison for all the datasets from LCBench regarding regret over the number of epochs. Similarly, in Figure 14 and 15, we show the same performance comparison, however, over time. As can be seen, DYHPO manages to outperform the other competitors in the majority of the datasets, and in the datasets that it does not, it is always close to the top-performing method, and the difference between methods is marginal.

In Figure 16 we provide the extended results of **Experiment 3** for TaskSet. We show the precision, average regret, and promotion percentage for poor-performing configurations for DYHPO and the other competitor methods.

Lastly, we explore the behavior of DYHPO after finding the configuration which is returned at the end of the optimization as the best configuration. In Figure 17, we show how the budget is distributed on the configurations considered during that part of the optimization. Clearly, DYHPO is spending very little budget on most configurations. Furthermore, we investigated how many new configurations are considered during this phase. For LCBench, 76.98% of considered configurations are new demonstrating that DYHPO is investigating most of the budget into exploration. These are even more extreme for TaskSet (93.16% and NAS-Bench-201 (97.51%).

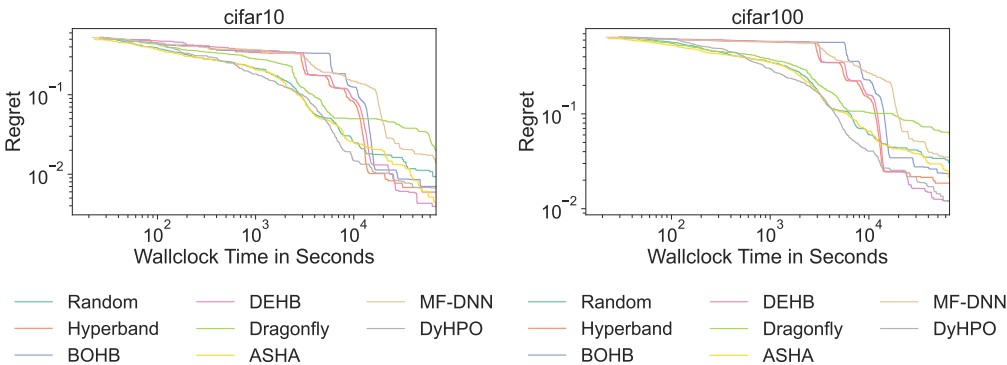

Figure 10: NAS-Bench-201 regret results over the total optimization time. The total time includes the method overhead time and the hyperparameter configuration evaluation time.

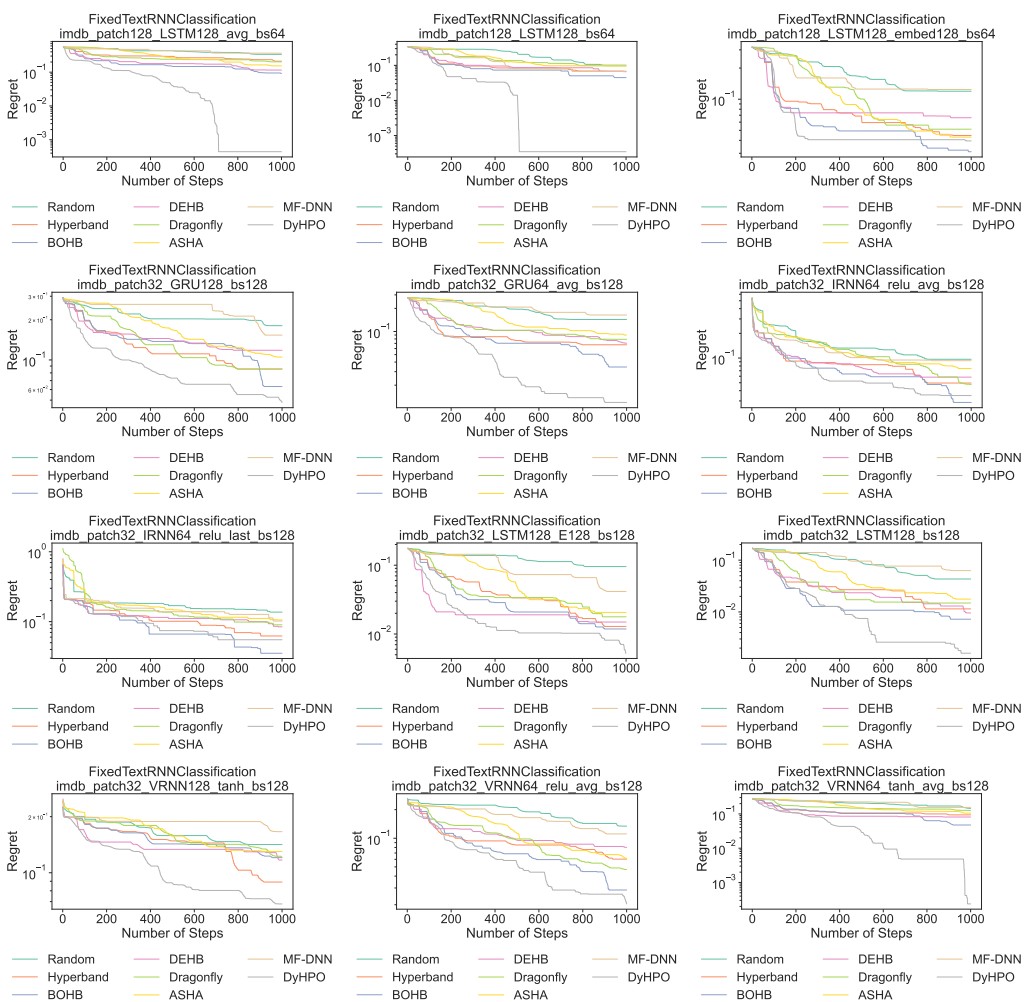

Figure 11: Performance comparison over the number of steps on a dataset level for TaskSet.

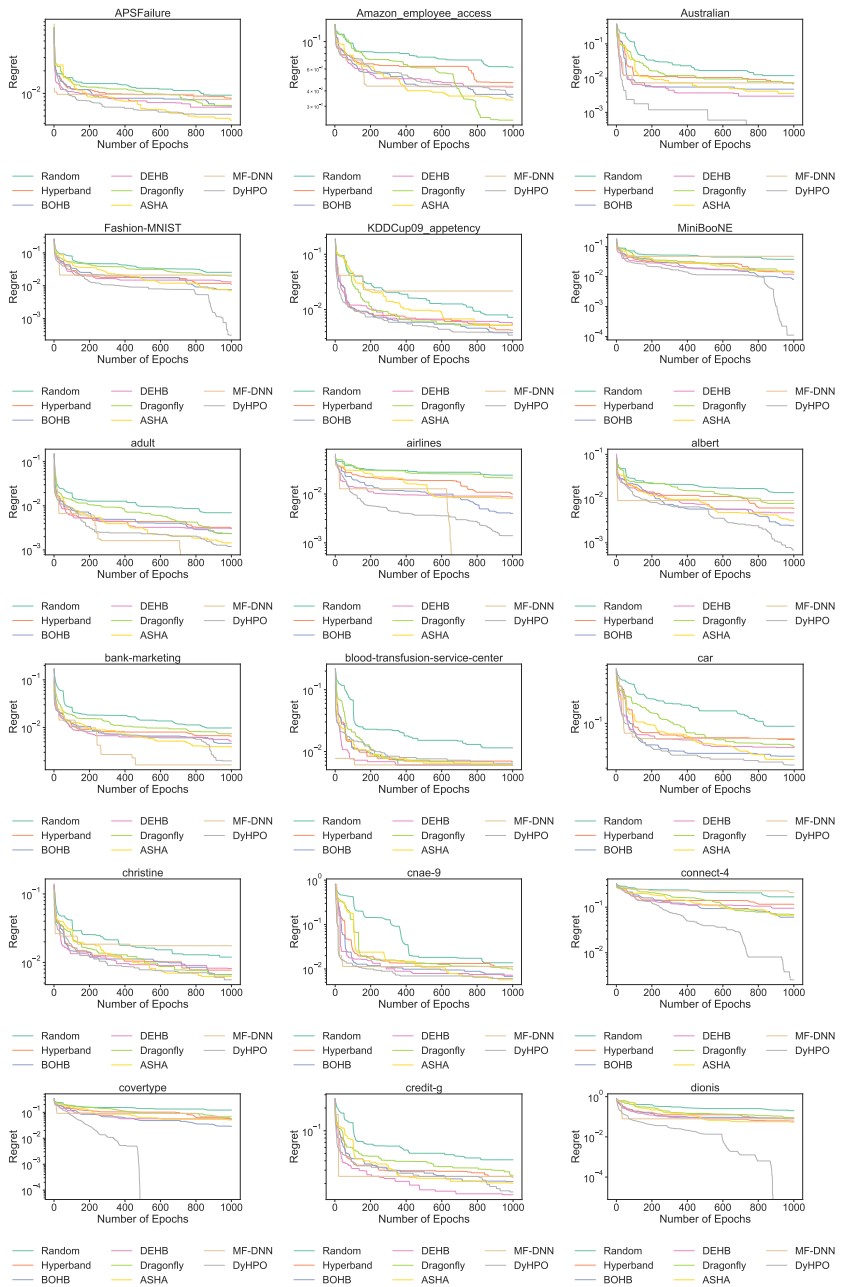

Figure 12: Performance comparison over the number of steps on a dataset level for LCBench.

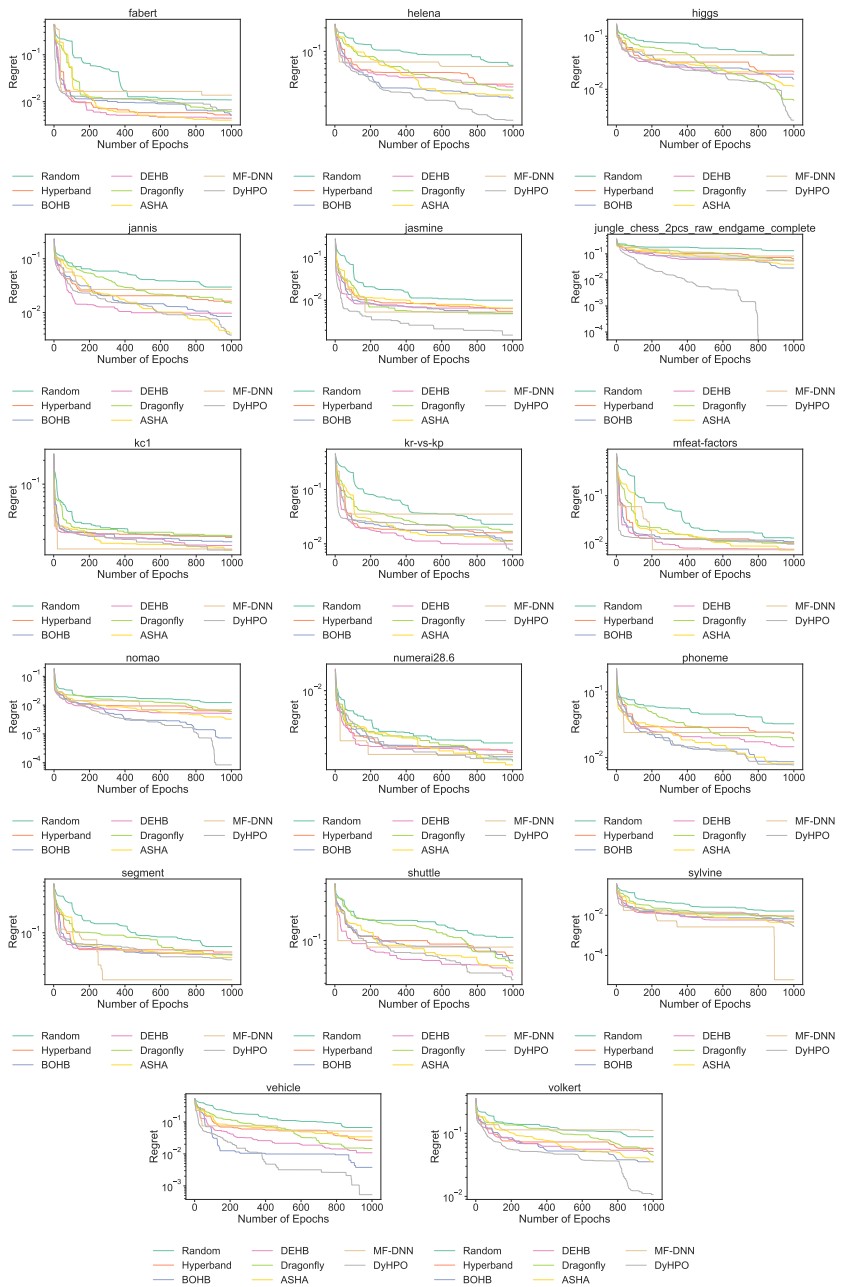

Figure 13: Performance comparison over the number of steps on a dataset level for LCBench (cont.).

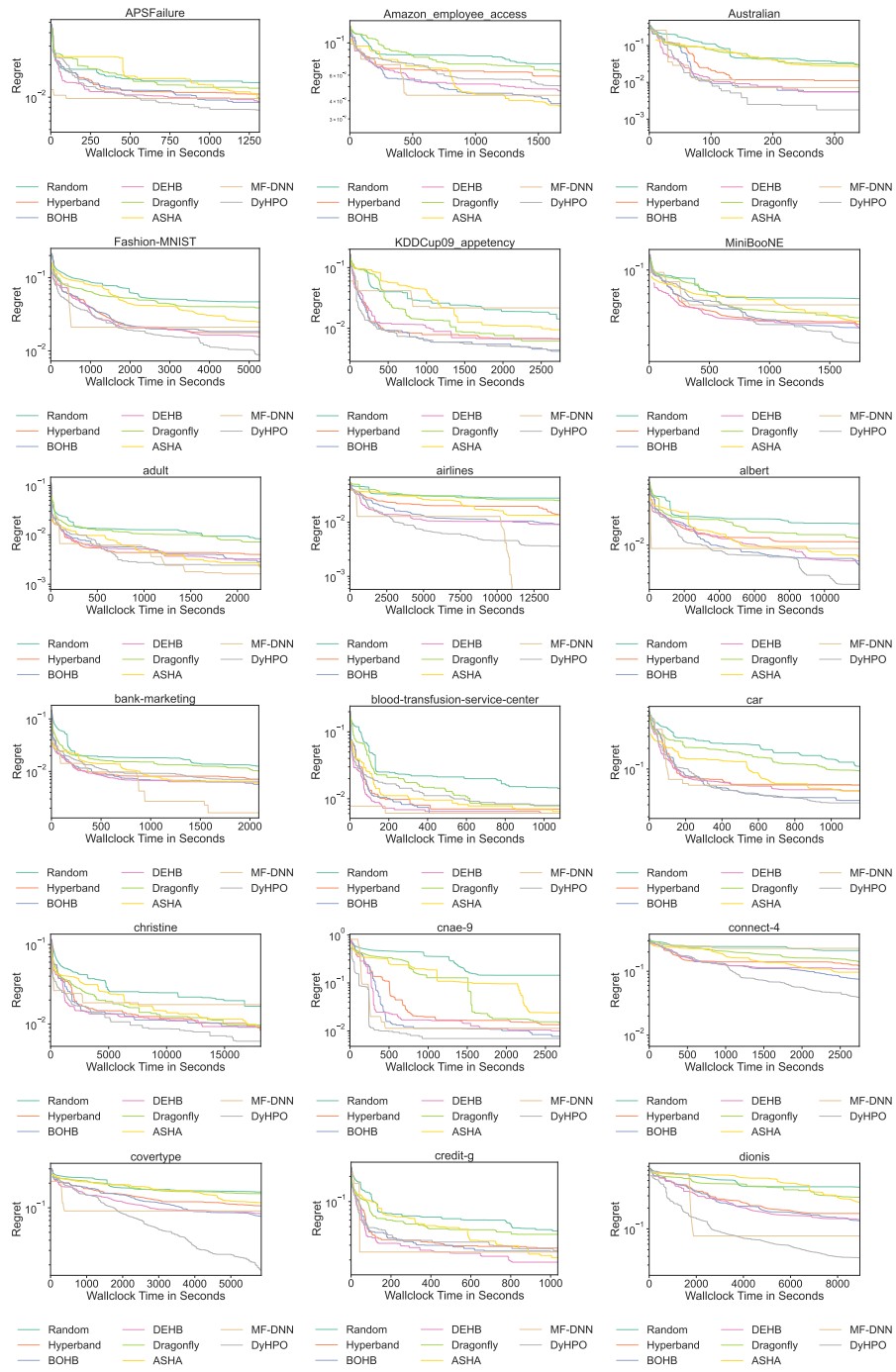

Figure 14: Performance comparison over time on a dataset level for LCBench with the overhead included.

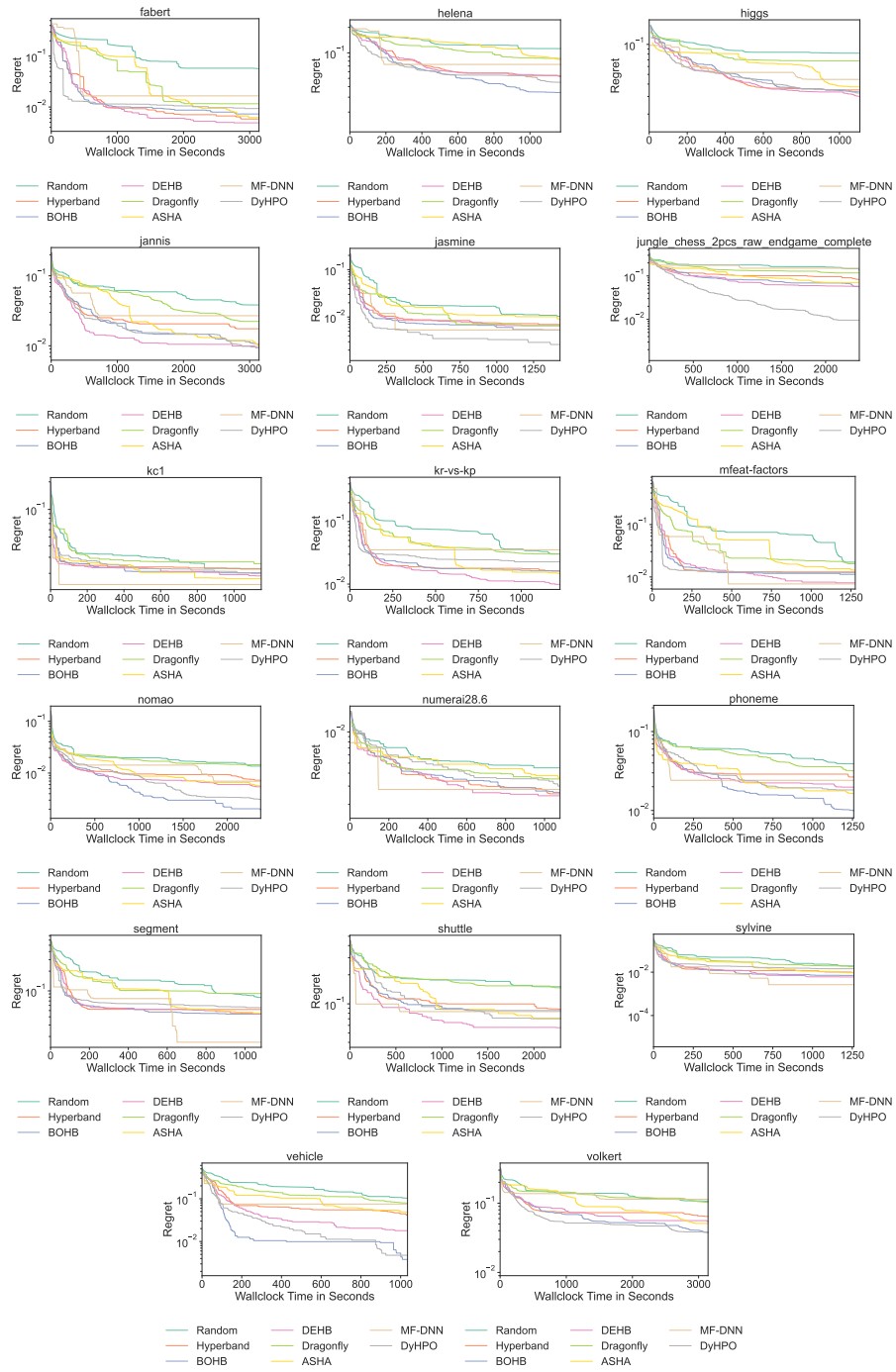

Figure 15: Performance comparison over time on a dataset level for LCBench with the overhead included. (cont.).

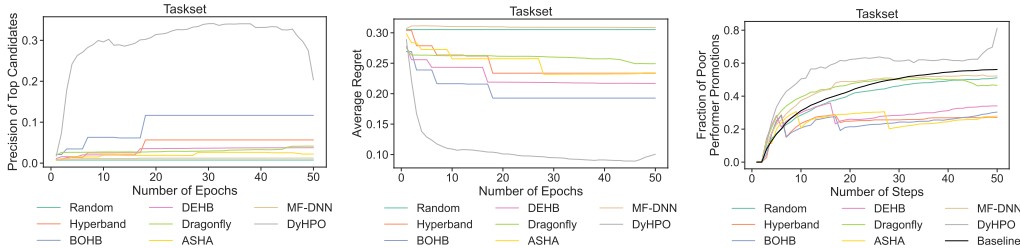

Figure 16: The efficiency of DYHPO as the optimization progresses. **Left:** The fraction of top-performing candidates from all candidates that were selected to be trained. **Middle:** The average regret for the configurations that were selected to be trained at a given budget. **Right:** The percentage of configurations that belong to the top 1/3 configurations at a given budget and that were in the top bottom 2/3 of the configurations at a previous budget. All of the results are from the Taskset benchmark.

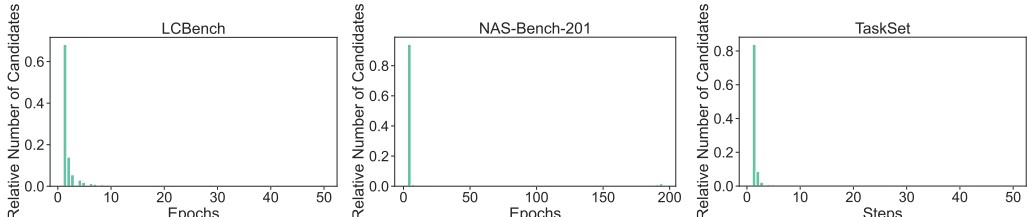

Figure 17: These plots shed light on how DYHPO behaves *after* the configuration it finally returns as the best. The plots show how many epochs are spent per candidate. As we can see, for most candidates only a small budget was considered, indicating that DYHPO is mostly exploring at this point.