# OpenReview forum: "Supervising the Multi-Fidelity Race of Hyperparameter Configurations"
_NeurIPS.cc/2022/Conference — NeurIPS 2022 Accept_

### Official Review · Reviewer_1bWR · 2022-07-10

**Rating:** 7
**Confidence:** 3
**Soundness:** 3 good
**Presentation:** 3 good
**Contribution:** 3 good

**Summary:**

The paper presents DyHPO, a new multi-fidelity method for hyper parameter optimisation. They key novelty at the basis of DyHPO is its new deep kernel GP. By relying on a deep kernel (based on a CNN) DyHPO can leverage the learning curve to recommend the next configuration to explore. DyHPO relies on an expected improvement based acquisition function to decide which configuration to recommend and evaluates each configuration at different fidelities, incrementally. The evaluation based on well-known datasets and state-of-the-art baselines shows that DyHPO achieves state-of-the-art results on the considered benchmarks, demonstrating its superiority and the advantage of its deep kernel.


**Questions:**

**a)** I would have liked to see a plot with the ablation line (DyHPO without CNN) and with the other baselines, because it is harder to evaluate whether DyHPO's gains are all due to the deep kernel (i.e., it is hard to compare DyHPO without the deep kernel with the baselines).

**b)** How does multi-fidelity EI contribute to the gains achieved by DyHPO? Did you try other acquisition functions? For example, [2] proposes an acquisition function based on entropy search (ES)

**c)** I assume "regret" is the same as "loss". If this is true perhaps you could clarify this in the text, as "loss" is a commonly used metric.

**d)** Why do you sometimes use "number of epochs" and others "number of steps" in the x-axis?

**e)** In some of the LCBench datasets (e.g., ppendix, figure 12, "Amazon_employee_access", "bank-marketing", "blood-transfusion-service-center", "credit-g") DyHPO is no longer as competitive with state-of-the-art methods. Do you have an intuition as to which special characteristics of these datasets may lead to a worse performance by DyHPO?

**f)** When you evaluate the precision (experiment 3, line 289), you explain your definition of precision (which is consistent with the "usual" definition of precision = TP / (TP + FP)) however in the plot the y-axis says true positives. Is this a typo or are you actually showing the true positives instead of the precision?

**g)** I'm assuming the budget is the number of epochs during which you evaluate a configuration (I don't think this is explicit in the text or I may have missed it) and also that you "maintain" a j for each configuration, and when that configuration is selected you increment it. Are these assumptions correct? This is not very clear in the text.

**h)** Given the previous assumptions, I wonder how many configurations are explored (and on how many budgets) when DyHPO has already found promising configurations. That is, once DyHPO finds the configuration it ends up recommending, how much more does it explore instead of exploit?

**i)** What is the total cost spent by DyHPO compared to the baselines?

**j)** The title of the submission is not very clear to me. Why "Supervising"?

**Comments:**
- maybe you could add some markers to each baseline to make it easier to distinguish each line. I had to zoom in quite a bit (e.g. on figure 3) to see the results easily.
- Figures 12 and 13: legend mentions steps, but x-axis has epochs

**Limitations:**

Given that the kernel is based on a CNN, which is more computationally expensive that "traditional" (non-deep) kernels, I would have liked to see a discussion on how relying on such a kernel may prevent DyHPO from being widely adopted (due to the compute power required to deploy it). Second, the problem of setting the parameters of the CNN is also not discussed. For example, if I were to use a shallower NN to reduce complexity, how would that impact performance? Similarly, what would the performance gains be if I made the network even deeper?

**Strengths And Weaknesses:**

**Originality:** The paper presents a new method for for hyper parameter optimisation, which is a novel combination of known techniques. The main difference from existing work is the fact that it relies on a deep kernel. Related work is adequately cite, [1] and [2] may also be of interest to the authors. [1] presents a new HPO approach based on HyperBand, while the one presented in [2] is a multi-fidelity approach that creates one model that exploits all fidelities, evaluating a configuration on a specific fidelity as a single "entity".

**Quality:** The claims are well supported, the evaluation methods and baselines are appropriate. The limitations of the proposed work are also discussed.

**Clarity:** The paper is clearly written and well structured. It's easy to follow and mostly easy to understand. It could do a better job at explaining how j is maintained and updated for each configuration.

**Significance:** The results are important as hyper parameter optimisation is an active area of research and given the demonstrated results of the proposed method, it is likely to have a high impact on the community. This is especially true if the source code is made available (as the authors say they will do).

[1] Mendes, Pedro, Maria Casimiro, and Paolo Romano. "HyperJump: Accelerating HyperBand via Risk Modelling." arXiv preprint arXiv:2108.02479 (2021).

[2] Mendes, Pedro, et al. "Trimtuner: Efficient optimization of machine learning jobs in the cloud via sub-sampling." 2020 28th International Symposium on Modeling, Analysis, and Simulation of Computer and Telecommunication Systems (MASCOTS). IEEE, 2020.

---

> ### Author Response · Authors · 2022-08-02
> **Response to Reviewer 1bWR**
>
> **On the ablation of the learning curve (CNN):** As suggested by the reviewer, we enhanced Figure 7 in Section 5 and Figure 8 in Appendix B to showcase the impact of the learning curve by comparing against the simplest baseline random search. As can be seen, DyHPO without the learning curve as input does not outperform random search.
>
> **On the ablation of the acquisition function:** To avoid being repetitive, please find above the answer to reviewer NJxE.
> We will extend the related work to discuss other acquisition functions and discuss the two works by Mendes et al. (for now in Appendix A.6).
>
> **On the regret term:** We refer the reviewer to Section 4.1 regarding the experimental setup, where we have clarified what the term regret signifies. We stress that regret is a standard metric in the HPO community and is used by most papers.
>
> **On the number of epochs vs number of steps:** We refer the reviewer to Section 4.4 to the description of Experiment 1 (additionally to the caption of Figure 3) where we have already described the difference between steps for TaskSet and epochs for the other benchmarks.
>
> **On the bad performance datasets:** Our interpretation so far is that the behavior is a limitation of Bayesian Optimization and Gaussian Process uncertainties. The uncertainty estimation of the performance of the first epoch for some yet unexplored hyperparameter configurations is small, and the acquisition function of these configurations remains a small value. Therefore, Bayesian Optimization does not explore a lot in some datasets, and instead exploits/advances known configurations. It is possible to control the acquisition function and parametrize the degree of exploration for avoiding this over-exploitation phenomenon, however, for simplicity, we wanted to have a single setting for all experiments, and not fine-tune our method for every dataset.
>
> **On the number of true positives:** We thank the reviewer for the feedback, the y-axis label does have a typo. We edited to “Precision of Top Candidates” in the revised version of our work.
>
> **On the budget and the value of j:**  The reviewer is correct in his understanding. We made it clearer in the updated paper version by revising Algorithm 1 for better clarity.
>
> **On the hyperparameters evaluated after the optimal one:** We refer the reviewer to the last paragraph of Appendix B. We show the distribution of budgets invested on different configurations in Figure 17 after finding the optimal one. Furthermore, we state how many of the configurations considered are novel.
>
> **On the DyHPO cost compared to the baselines:** We kindly refer the reviewer to the response for reviewer XM9p.
>
> **On the title:** We also propose “Efficient Multi-fidelity Hyper-Parameter Optimization for Deep Learning” if the reviewer thinks this is a better description. Otherwise, we welcome any suggestions from the reviewers.
>
> **On the limitations:** As mentioned in Appendix A.4 our neural network consists of only 2 fully-connected layers and 1 convolutional layer. Furthermore, DyHPO converges faster than all the other baselines, which indicates that the overhead is not significant and does not impact the results. Lastly, we would like to mention that the actual settings already lead to state-of-the-art results and further tuning of the experiments could only make our method stronger.
>
> If that answers the reviewer’s concern we invite the reviewer to reflect our feedback to the concern in the score.

---

> > ### Comment · Reviewer_1bWR · 2022-08-08
> > **Thank you for the clarifications**
> >
> > I thank the authors for the clarifications.
> >
> > Regarding the title of the submission, I do not request a change in the title.
> > I was merely enquiring after the rationale behind the current title.
> >
> > I have no further questions, thank you.

---

### Official Review · Reviewer_XM9p · 2022-07-11

**Rating:** 7
**Confidence:** 4
**Soundness:** 3 good
**Presentation:** 3 good
**Contribution:** 3 good

**Summary:**

This paper proposed a novel multi-fidelity grey-box Bayesian optimization method based on deep kernel GP.

**Questions:**

Could the authors discuss more the efficiency in terms of time?

**Strengths And Weaknesses:**

Pros

The motivation is clear and well-defined. Good summary of related work.  The experiments are comprehensive. Limitations were also discussed.

---

> ### Author Response · Authors · 2022-08-02
> **Response to Reviewer XM9p**
>
> **On the efficiency regarding time:** We are glad that the reviewer finds our work interesting and well-written.  We would like to point out the reviewer to Hypothesis 2 at Section 4.4 and to Section 5, Experiment 2, where we discuss DyHPO’s overhead and conclude that the effect of the overhead is negligible on the results and we still outperform the other baselines.
>
> If that answers the reviewer’s concern we invite the reviewer to reflect our feedback to the concern in the score.

---

> > ### Comment · Reviewer_XM9p · 2022-08-04
> > **Acknowledgement of response**
> >
> > Thanks for the detailed response from the authors. I appreciate the thorough response here.

---

### Official Review · Reviewer_MHNh · 2022-07-11

**Rating:** 8
**Confidence:** 3
**Soundness:** 4 excellent
**Presentation:** 3 good
**Contribution:** 4 excellent

**Summary:**

In the paper "Supervising the Multi-Fidelity Race of Hyperparameter Configurations" an HPO method for neural networks is proposed based on the framework of Bayesian optimization. To this end, the authors propose a kernel function, which is mainly implemented via a neural network, that accounts for different fidelities and is thus able to make use of all the data in contrast to other methods that employ a distinct surrogate model for each fidelity level.

**Questions:**

* ll.16ff: What about SMAC? Which is a method particularly tweaked towards the configuration of algorithms and therewith also for HPO?
* What are the drawbacks and limitations of DyHPO?

**Limitations:**

Conceptual limitations are not discussed. However, the scope of the proposed HPO method is more or less clearly limited to the hyperparameter optimization for neural networks.

**Strengths And Weaknesses:**

## Strengths
The authors address a severe issue of current multi-fidelity methods which are prone to reject/cancel out hyperparameter configurations which perform bad on early budget stages but would outperform configurations promoted to the next multi-fidelity levels on higher budgets. As an example the authors explicitly mention neural networks which are trained with regularization and known to perform better in the long run but are dominated by configurations that are not regularizing for lower budgets. I agree with the authors that this problem is one of the main drawbacks of multi-fidelity methods. Another example would be to compare more complex networks, which require more data for training, to simpler networks, maybe even linear ones, which perform sufficiently well on lower budgets already. In this case, the configurations representing the simpler candidates would seem beneficial in the first place and be promoted to subsequent stages.

The proposed kernel function seems to me like a reasonable and original contribution. At least I am not aware of any such method which would pursue a very similar strategy which is not discussed in the paper.

In the results the proposed approach is found to yield state-of-the-art performance. More specifically, DyHPO shows outstanding performance.

Overall, the paper is clearly written and well structured. Especially the experimental part, I really enjoyed as it was well structured and the addressed research questions have been made very explicit.

## Weaknesses
The authors promise to release their code upon acceptance. However, for a thorough review process it would be more appropriate to also hand in the source code.

Figures and algrotihms are positioned inconveniently throughout the document. First, text floats around some of the figures, I assume to save some space. Honestly speaking, I do not know whether this is against the style guidelines. In a hurry, I could not find anything about this in the guidelines, so I assume it is okay, still I would consider it a bad practice and "cheating on the space limitations" to some extent. Figures should be aligned to the top to avoid outcomes as on page 7 where two lines of text are given above the figure. As a reader I did not expect that and first missed the two lines. Also the fonts in the figures are sometimes too small and thus hard to read as a printed version.

---

> ### Author Response · Authors · 2022-08-02
> **Reply to Reviewer MHNh**
>
> **On the code release:** To avoid being repetitive, please find above the clarification to reviewer NJxE.
>
> **On the comparison with SMAC**: SMAC is by design not a multi-fidelity HPO method in its core. It is true that there exists a recent combination of SMAC with Hyperband (https://automl.github.io/SMAC3/master/apidoc/smac.intensification.hyperband.html), however, we already have Hyperband and two recent variations BOHB, DEHB as baselines.
>
> **On the limitations:** We agree with the reviewer that the pause and resume part of our procedure is viable when tuning the hyperparameters of parametric models only, however, in the case of non-parametric algorithms, one could start training from the beginning and evaluate every k steps and not every step, which in turn would translate into a fixed schedule.
>
> We thank the reviewer for the feedback and we have updated the limitations section in the revised version of our work following the suggestions from all the reviewers.
>
> **On the Figures and Font Size:** We appreciate the feedback and we improved readability accordingly.
>
> If that answers the reviewer’s concern we invite the reviewer to reflect our feedback to the concern in the score.

---

> > ### Comment · Reviewer_MHNh · 2022-08-05
> > **Acknowledgement of response**
> >
> > Thanks for the target-oriented response and the clarifications.

---

### Official Review · Reviewer_NJxE · 2022-07-13

**Rating:** 6
**Confidence:** 4
**Soundness:** 3 good
**Presentation:** 3 good
**Contribution:** 3 good

**Summary:**

The authors consider multi-fidelity hyperparameter optimization (HPO). They introduce a new method, DyHPO, which uses Bayesian optimization (BO) with a newly proposed deep kernel that takes as input the full learning curve. Their approach also uses a new multi-fidelity version of expected improvement. At every iteration, DyHPO decides which hyper parameter set to train for an additional fidelity (epoch). The authors run extensive experiments on 50 datasets spanning tabular, image, and NLP data.

**Questions:**

The authors say the will release the code in the camera ready version. Can the authors release their code during the author response period? For example, by using https://anonymous.4open.science/?  This would improve the contribution.

Can the authors run more ablation studies, such as some of the ones mentioned in the weaknesses?

Can the authors comment on Freeze-thaw Bayesian optimization and ideally run empirical comparisons?

**Limitations:**

The authors do include limitations and societal impact, but this paper is technically over the page limit, since the societal impact section is supposed to fit within the first 9 pages. The authors should get the societal impact section into the first 9 pages.

**Strengths And Weaknesses:**

### Strengths
This is a clean method that performs well. Intuitively, methods such as Hyperband or BOHB could make use of learning curve extrapolation to improve performance, so it is great to see some progress in this direction.

Extensive experiments: the diversity and the quantity of the experiments is well-done. They chose seven other benchmarks, from baselines to state-of-the-art techniques.

They have evaluated three research questions by giving hypotheses and then designing experiments to test the hypotheses.

### Weaknesses
It is not particularly novel. All of the pieces are similar to existing work (e.g., multi-fidelity Bayesian optimization, deep kernel learning, freeze-thaw Bayesian optimization, and learning curve extrapolation work). However, it is still non-trivial to design this method.

The approach seems pretty similar to Swersky et al. 2014 (freeze-thaw Bayesian optimization). That approach also uses Bayesian optimization to decide whether to pause or continue training dynamically over all hyperparameter settings. (The difference is deep kernels, using the full learning curve, and the specific acquisition function.) Ideally, the authors should compare to this method in the experiments, given that it is so similar.

I would have liked to see more ablation studies. For example, the authors do not really mention some related works on extrapolating learning curves, such as the following:
- Speeding up automatic hyperparameter optimization of deep neural networks by extrapolation of learning curves
- Learning curve prediction with Bayesian neural networks
- Probabilistic Rollouts for Learning Curve Extrapolation Across Hyperparameter Settings
- Accelerating neural architecture search using performance prediction

The authors could discuss and compare to these other techniques that implicitly or explicitly use the full learning curve dynamics.

The authors could also run an ablation study on the acquisition function, and the deep kernel vs another surrogate.

Lines 241-242 seems strange because you can simply decrease the amount of fidelities. For example, instead of treating every epoch separately in DyHPO, you could only allow DyHPO to query epochs 5, 10, 15, 20, etc.

No code.

---

> ### Author Response · Authors · 2022-08-02
> **Response to Reviewer NJxE**
>
> **On Freeze-Thaw:** The reviewer is correct, it is a very relevant baseline. In fact, Freeze-Thaw is a seminal paper in the field of multi-fidelity HPO. Sadly, there is no code available for it and that is the reason why we (and the related work) regretfully skipped this baseline from the experiments. We did our best to add Freeze-Thaw and re-implemented it from scratch internally, but the original results could not be reproduced as several design details are lacking in the Freeze-Thaw paper. Our implementation led to poor results, i.e. slightly better than random search. Other domain expert colleagues who published papers on multi-fidelity HPO in the past had the same reproducibility concerns with Freeze-Thaw too. On the other hand, non-official public implementations are old and in our experiments (after spending days of fixing dependencies and library versions) they also performed close to random search. This is the story behind not having it as a baseline.
>
> **On the code release:** We are working on releasing the source code, which as promised will be public until the camera-ready version. The responsible first author comes from a research institution, where releasing the code has to pass through a long approval process (a norm for most non-university research labs). On the other hand, we have provided enough implementation details for the results to be reproduced, in the case that something is unclear, we are more than willing to clarify it further.
>
> We will double our efforts to polish the source code and release it even sooner than promised, hopefully during the rebuttal.
>
> **On related work:** Notice that our novelty is in having a probabilistic surrogate for the performance prediction at different budgets (a.k.a. learning curve extrapolation), which combines the uncertainty of GPs (as it is crucial for Bayesian Optimization) with the power of deep learning representations (incl. learning curves and budget information) with a deep kernel.
>
> The papers that the reviewer recommends do not use a GP-based surrogate:
>
> - Speeding up automatic hyperparameter optimization of deep neural networks by extrapolation of learning curves (2015)
> - Learning curve prediction with Bayesian neural networks (2016)
> - Probabilistic Rollouts for Learning Curve Extrapolation Across Hyperparameter Settings (2019)
> - Accelerating neural architecture search using performance prediction (2018)
>
> In that aspect, the recommended related work is loosely connected from a technical perspective to our method. They represent older baselines  (2016-2019) that use other types of multi-fidelity surrogates (random forests, parametric curve models, bayesian neural networks, MLPs) with learning curves. Regarding A, B, and C we used more recent and stronger multi-fidelity baselines from the same research group that published A, B, C, concretely BOHB 2018, DEHB 2021. Overall, we already compared against 7 baselines and we have 4 strong state-of-the-art baselines that are published >= 2020. We will cite all the recommended papers above in the updated related work (for now we have it in Appendix A.6). If the reviewer believes we missed a strong recent baseline, we would be happy to consider it.
>
> **On the ablation of the deep kernel vs another surrogate:**
>
> The demanded ablation of the surrogate is already in the results. We compared it to Dragonfly, a library implementing BOCA (Kandasamy 2020, JMLR), which uses a GP with a special multi-fidelity kernel.
>
> **On lines 241-242:**  We agree with the reviewer. In essence, different optimization settings are possible:
> I. update the surrogate every 1 learning curve epoch/fidelity, for N surrogate training steps (i.e. steps to update the deep kernel of a GP), or
> II. every K epochs/fidelity for N*K surrogate training steps.
>
> We decided to do more frequent optimizations after every learning curve epoch, but update the surrogate for fewer optimization steps. In the end, this is a matter of hyper-hyper-parameter tuning of our method. The actual settings lead to state-of-the-art results already, and intensive fine-tuning of our settings can only make our method stronger empirically. To promote simplicity and reproducibility, we opted for having a single set of settings for all experiments of the paper (i.e. always query the next epoch), and not choose different fidelity frequencies for different benchmarks.
>
> **On the limitations section:** We have updated the limitations section in the revised version of our work following the suggestions from all the reviewers.
>
> If that answers the reviewer’s concern we invite the reviewer to reflect our feedback to the concern in the score.

---

> > ### Comment · Reviewer_NJxE · 2022-08-08
> > **Thank you for your response**
> >
> > I thank the authors for addressing my concerns. I have a few follow-up comments and questions.
> >
> > **Freeze-thaw** Thank you for the explanation.
> >
> > **Code release** I believe this work would have higher impact if the code is released and easy to use.
> >
> > **Related work**
> > My main point is that there are currently **no** comparisons to other methods that do explicit learning curve extrapolation. Your current baselines include only multi-fidelity algorithms such as BOHB, DEHB, and Dragonfly.
> >
> > Thus, it seems natural to include at least one or two learning curve extrapolation methods as a baseline to your method, such as the four I mentioned. In my original review, my thought was to include some of these methods in ablation studies, for example, if there’s a way to fix the outer framework (the search procedure) and run an ablation on the surrogate (the part that predicts learning curves / configurations). Now I think this could be nontrivial. But it still seems natural at least to do an end-to-end comparison with some of the LCE methods, even though they are from 2019 or earlier.
> >
> > **Ablation studies on surrogate and on aquisition function**
> > What I meant was not an end-to-end comparison, but an ablation study. The goal of such an experiment would be to see whether the deep kernel or the acquisition function (or both) was the reason DyHPO performs well. It sounds like Reviewer 1bWR had the same questions with their question a) and b).
> >
> > Comparing to Dragonfly does not answer these questions, because although it’s similar, I am sure that the outer BO loop is not identical to DyHPO. It would be better to run a true ablation study where everything except the surrogate and/or acquisition function is fixed as equal.
> >
> > **Lines 241-242, limitations section, and updated related work in the appendix** All of these look like great additions.

---

> > > ### Author Response · Authors · 2022-08-09
> > > **Thanks for the suggestions**
> > >
> > > We thank the reviewer for the precious recommendations on conducting further analyses, which we will gladly consider for our future follow-up work.

---

### Author Response · Authors · 2022-08-09
**DyHPO Code**

As promised we are releasing our code base before the promised time frame (i.e. before the camera ready deadline). Additionally, we will continuously polish and keep documenting the code in the upcoming days. When the reviewing process is completed, we make the code available in a public repository.

Anonymous repository: https://anonymous.4open.science/r/DyHPO-7D76/README.md

---

### Meta-Review · Area_Chair_vZmV · 2022-08-24

**Recommendation:** Accept
**Confidence:** Certain

**Metareview:**

All reviewers consider this paper to make a good contribution to multi-fidelity BO, it is "a novel combination of known techniques", and achieves state-of-the-art performance. Multiple reviewers ask for releasing code which authors promise to do before camera-ready deadline. The authors' feedback also addresses most other concerns or questions from the reviewers.

After discussion, the remaining concerns from one reviewer are that (1) authors should discuss more on related works that conduct learning curve extrapolation (LCE) and make a comparison to one of prior works (2) lack of ablation studies to answer e.g., "why does DyHPO achieve strong performance: the deep kernel, or the mulit-fidelity EI acquisition function? For (1), it would be useful to include the discussion on LCE related work from Appendix A.6 and the reply to NJxE into the main body of the paper. For (2), another reviewer finds the enhanced Figure 7 in Section 5 and Figure 8 in Appendix B from rebuttal useful to showcase the impact of the learning curve. I would encourage the authors to take into consideration the remaining concerns from the reviewers and edit their final revision accordingly.

**Award:**

No

---

### Decision · Program_Chairs · 2022-09-14

Accept